# Low-Temperature Hydrothermal Synthesis of Chromian Spinel from Fe-Cr Hydroxides Using a Flow-Through Reactor

Yoko Ohtomo [1,*], Jeehyun Yang [1,†], Miu Nishikata [2], Daisuke Kawamoto [3], Yuki Kimura [4], Tsubasa Otake [1] and Tsutomu Sato [1]

[1] School of Engineering, Hokkaido University, Kita-13, Nishi-8, Sapporo 060-8628, Japan
[2] Geological Survey of Japan, National Institute of Advanced Industrial Science and Technology, 1-1-1 Higashi, Tsukuba 305-8567, Japan
[3] Faculty of Science, Department of Chemistry, Okayama University of Science, 1-1 Ridaicho, Kita-ku, Okayama 700-0005, Japan
[4] Institute of Low Temperature Science, Hokkaido University, Kita-19, Nishi-8, Kita-ku, Sapporo 060-0819, Japan
* Correspondence: ohtomoy@eng.hokudai.ac.jp
† Current address: Jet Propulsion Laboratory, California Institute of Technology, National Aeronautics and Space Administration, 4800 Oak Grove Dr, Pasadena, CA 91109, USA.

**Abstract:** Recent studies have suggested that a chromian spinel can be formed under natural hydrothermal conditions; however, the required conditions, process, and associated redistribution of Cr are still poorly understood. Here, chromian spinel formation was performed by Fe-Cr hydroxides alteration with an $Fe^{2+}_{(aq)}$ supply at 150, 170, and 200 °C and 5 MPa simulating the diagenetic process. The flow-through system enabled the $Fe^{2+}_{(aq)}$ supply to be leached from the magnetite by an acidic solution to synthesize Fe-Cr hydroxides as the starting material with two reaction cells, flow lines, heaters, and a high-performance liquid chromatography (HPLC) pump. The accuracy of the temperature measurement was confirmed based on the amorphous silica solubility. Mineralogical analysis of solid samples recovered from the reaction cell indicated that the chromian spinel was formed between 150 and 170 °C from Fe-Cr hydroxides through goethite with a simultaneous hematite formation, while Mössbauer spectra showed that a large quantity of Fe-Cr ferrihydrites still remained after the experiments probably because of the Cr addition to the stability of ferrihydrites. The Cr/Fe ratio of the chromian spinel was smaller than that of the bulk of the Fe-Cr ferrihydrites and equivalent to Cr-rich magnetite, suggesting a redistribution of Cr during the transformation from goethite to synthesized spinel under the hydrothermal conditions.

**Keywords:** chromian spinel; ferrihydrite; goethite; hydrothermal alteration; flow-through reactor

## 1. Introduction

Chromian spinel has been recognized as a typical high-temperature igneous mineral recording magmatic process in nature [1–6]. Recent petrological investigations have suggested that a chromian spinel could be nucleated from hydrothermal fluids during metamorphic or metasomatic events at lower temperatures [7–9]. On the other hand, it also has been interpreted as relics of igneous phases [10]. The existence of hydrothermal chromian spinel formation in nature is therefore still controversial. In previous experimental studies, chromian spinel formation under hydrothermal conditions (ca. <300 °C) has been performed using a batch-type pressure vessel in a pioneering experiment in the engineering field [11]; however, such closed system equipment cannot demonstrate chromian spinel formation in a natural, oxygen-free hydrothermal system with fluid-circulation. When assuming the formation in a natural hydrothermal system, most of the possible precursors for chromian spinel are Fe-Cr hydroxides because $Fe^{2+}_{(aq)}$ in a hydrothermal fluid is oxidized when mixed with dissolved $Cr^{6+}$ by the difference in the redox potential, resulting in precipitation of Fe-Cr hydroxides. Then the reduction of Fe-Cr hydroxides is

required to form chromian spinel in classic thermodynamic calculation at equilibrium. However, a previous experimental study has indicated that the non-redox transformation of Fe-bearing minerals could occur under $H_2$-rich acidic hydrothermal conditions via dissolution and reprecipitation as shown in this reaction [12]:

$$Fe_2O_3 + Fe^{2+}_{(aq)} + H_2O = Fe_3O_4 + 2H^+ \tag{1}$$

Reaction (1) is reversible, and redox-independent replacement of magnetite by hematite has been experimentally investigated and widely observed in natural iron ores in recent studies [13–15]. This finding led us to the idea that a chromian spinel can also be formed by a similar non-redox acid-base reaction in low-temperature hydrothermal conditions. In this study, we developed a high-pressure flow-through reactor to conduct hydrothermal chromian spinel synthesis from Fe-Cr hydroxides with an aqueous ferrous iron ($Fe^{2+}_{(aq)}$) supply under mildly acidic conditions (pH of 3–5) with the assumption of oxygen-free hydrothermal environments. The flow-through experiment was performed at various solid-liquid ratios, with various initial pHs of the supplied solutions, and various flow rates to examine the physicochemical conditions, especially the lower limit of temperature, feasible for chromian spinel formation at 150, 170, and 200 °C.

## 2. Materials and Methods

Fe-Cr hydroxides were synthesized as the starting material in this experimental study. The 0.05 mol/dm$^3$ $Na_2CrO_4$ and 0.05 mol/dm$^3$ $FeCl_2$ solutions were prepared by dissolving $Na_2CrO_4 \cdot 4H_2O$ (sodium chromate tetrahydrate; Kanto Chemical Co., Inc., Tokyo, Japan) and $FeCl_2 \cdot 4H_2O$ (iron (II) chloride tetrahydrate; Wako, Osaka Japan) in ultrapure water. These two solutions were mixed at Cr/Fe ratios of 0.2 to 0.33, and the pH was adjusted to ca. 7 using 1–10 mol/dm$^3$ NaOH solution. Brown precipitates of Fe-Cr hydroxides were immediately formed after mixing the two solutions. After the solution was mixed for 1 h and the pH became stable, the Fe-Cr hydroxide precipitates were separated, washed three times with ultrapure water by centrifugation (1650 g, 40 min), and then freeze-dried. The synthesized precipitates were analyzed using Fourier transform infrared spectroscopy (FT-IR; FT/IR-4100, JASCO, Tokyo, Japan) and were identified as Fe-Cr hydroxides [16].

The flow-through reactor comprised two stainless steel (SUS316) reaction cells (152 mm long, 20 mm diameter, 47.7 cm$^3$ maximum inner volume) connected with a single flow line (SUS316) to enable high temperature and pressure experiments (up to 200 °C and 10 MPa; Figure 1). Porous weld frits (SUS316, 2 μm) were placed at both ends of the cells to contain solid particles within the cells. The system pressure was controlled by a needle-type back pressure valve (1764-22, TESCOM, Emerson Electronic Co., St. Louis, MO, USA) and monitored before, after, and between the cells. A safety valve was set in the line between the high-performance liquid chromatography (HPLC) pump (PU-2080, JASCO, Tokyo, Japan) and Cell 1. The pressure differences were typically less than 0.1 MPa so that clogging did not occur in the cells during experiments. The cells and line were heated by a mantle heater (the red dot–dash squares in Figure 1) and ribbon heaters (yellow dot–dash squares in Figure 1), respectively, for temperature control. A series of preliminary experiments using a thermocouple to directly measure the solution's temperature in Cell 2 showed a significant difference (ca. 15 °C) between the internal solution temperature (in the middle of Cell 2) and the controlled temperature outside the cells (inside the mantle heater), depending on the set temperature and flow rate. The set temperatures for the flow-through experiments were calculated based on the difference from the controlled temperature. The accuracy of the temperature measurement was examined by measuring the dissolved amorphous silica concentration (amorphous silica C-200, Wako, Osaka, Japan) inside Cell 2 before experiments for a hydrothermal chromian spinel since the solubility has been well known for temperature dependency [17].

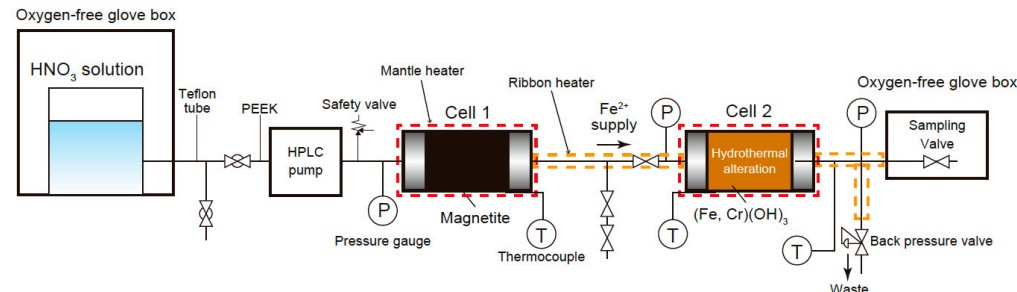

**Figure 1.** Schematic illustration of the flow-through reactor system used for hydrothermal chromian spinel synthesis (<200 °C, <10 MPa).

In hydrothermal chromian spinel synthesis, a $HNO_3$ solution of initial pH 3–4 (±<0.03) was prepared at room temperature by diluting 60% of a $HNO_3$ solution (Kanto Chemical Co., Inc., Tokyo, Japan) with boiled ultrapure water in an oxygen-free glove box ($O_2$ < 0.01%) that was bubbled by $N_2$ gas until the dissolved oxygen concentration reached less than 0.01 ppm to prevent $Fe^{2+}_{(aq)}$ oxidation during the experiment. The ionic strength of the solutions was adjusted to ca. 0.1 by the addition of $NaNO_3$ (Kanto Chemical Co., Inc., Tokyo, Japan). The solution was then introduced by the HPLC pump into Cell 1 filled with ca. 7–35 g of magnetite (1 μm, 99.9%, Kojundo Chemical Laboratory Co., Ltd., Saitama, Japan) to leach $Fe^{2+}_{(aq)}$ from the magnetite based on the following reaction [12]:

$$Fe_3O_4 + 2H^+ = Fe_2O_3 + Fe^{2+}_{(aq)} + H_2O \tag{2}$$

Note that reaction (2) was the inverse of reaction (1). After the reaction with magnetite in Cell 1, the solution was supplied to Cell 2, which was filled with Fe-Cr hydroxides. The temperature, pressure, and flow rate for the alteration experiments were set at 150–200 °C, 5 MPa, and 1–2 $cm^3$/min, respectively. Both Cells 1 and 2 were set at the same temperature. The internal solution temperature was monitored and recorded in the experimental runs. The experiments were conducted for 2 to 7 days. The experimental conditions for runs 1–4 are summarized in Table 1. The initial pH of the $HNO_3$ solution and temperature were set at 2.97–4.00 and 150–200 °C, respectively, assuming mildly acidic hydrothermal conditions. Fe-Cr hydroxides were synthesized at the Cr/Fe ratio of 0.25 according to the method from [16]. The synthesis of homogeneous Cr-rich starting material was difficult due to the rapid precipitation of Cr hydroxides under atmospheric conditions; therefore, we varied the Cr/Fe ratios in the starting materials in a narrow range from 0.2–0.33. The flow rate was set to 1–2 $cm^3$/min to minimize the temperature differences within the cells (<10 °C). The experimental durations were 2–7 days. The duration of the experiment was recorded from the point when the internal temperature of reaction Cell 2 reached the set temperature.

**Table 1.** Experimental conditions for hydrothermal chromian spinel formation by hydrothermal alteration of Fe-Cr hydroxides.

| Run | Initial pH | Cr/Fe Ratio of Starting Material | Amount of Magnetite | Amount of Starting Material | Flow Rate | Experimental Duration | Set Temperature |
|-----|-----------|----------------------------------|---------------------|----------------------------|-----------|----------------------|-----------------|
|     |           |                                  | (g)                 | (g)                        | ($cm^3$/min) | (day)             | (°C)            |
| 1   | 2.97      | 0.25                             | 35                  | 1.214                      | 2.0       | 2                    | 170             |
| 2   | 4.00      | 0.20                             | 20                  | 1.010                      | 2.0       | 5                    | 150             |
| 3   | 3.00      | 0.25                             | 20                  | 1.335                      | 1.0       | 7                    | 150             |
| 4   | 3.00      | 0.33                             | 7                   | 8.115                      | 1.0       | 7                    | 200             |

The solution passing through Cell 2 was periodically sampled (10–20 $cm^3$) from a back pressure valve (1764-22, TESCOM, Emerson Electronic Co., MO, USA) in the oxygen-free glove box ($O_2$ < 0.5%) in each run. The oxygen-free glove box was repeatedly evacuated and

purged with $N_2$ before solution sampling until the $O_2$ volume sufficiently decreased. The pH of the solution samples was measured with a pH meter (LAQUA F-72, Horiba, Kyoto, Japan) at room temperature. The solution samples were then filtered by a 0.2 μm membrane, acidified with 10 mol/dm$^3$ $HNO_3$ solution, and analyzed with respect to the dissolved Fe concentration using inductively coupled plasma–atomic emission spectroscopy (ICP–AES; ICPE-9000, Shimadzu, Japan). The detection limit of the Fe concentration for the ICP-AES analysis was ca. 0.1 ppm. Dissolved silica concentrations were measured using UV-vis spectrophotometry (UV/VIS spectrophotometer V-550, JASCO, Tokyo, Japan) based on the formation of blue molybdesilisic acid for temperature correction experiments using silica solubility [18].

After the experiments, solid samples recovered from Cell 2 were dried at room temperature and analyzed using powder X-ray diffraction (XRD; XRD Multiflex, Rigaku, Japan) with graphite-monochromatic incident Cu Kα radiation (40 kV and 30 mA) to identify constituent minerals. In the XRD analysis, samples were scanned from 5–70° (2θ) with a scanning speed of 2.0°/min. Peaks in obtained profiles were identified using the database of Crystal Maker 9.0 (Crystal Maker Software Ltd., Oxfordshire, UK). To identify Fe-bearing phases, Mössbauer spectroscopy was performed on samples diluted with agarose and reference materials (α-Fe, $FeSO_4 \cdot 7H_2O$, goethite, magnetite, and chromite) using $^{57}Co$ source γ-rays (Radioisotope Center, Kyushu University, Fukuoka, Japan). The zero velocity position of the spectrum was the center of gravity for the α-Fe foil. Obtained spectra were deconvoluted and analyzed using Origin-Pro 2018 software (OriginLab, Northampton, MA, USA). Powder samples were directly installed on silicon wafers for observation using field emission-scanning electron microscopy-energy dispersive X-ray spectroscopy (FE-SEM-EDS; JSM-6500F, JEOL, Tokyo, Japan). Powder samples were also suspended in 70% ethanol using an ultrasonic homogenizer, dropped on microgrids, and observed using transmission electron microscopy-energy dispersive X-ray spectroscopy (TEM-EDS; JEM-2100F, JEOL, Tokyo, Japan). Obtained diffraction data were identified using Single Crystal and the database of Crystal Maker 9.0.

## 3. Results and Discussion

### 3.1. Temperature Measurements Using Amorphous Silica Solubility

Most of the dissolved silica concentrations measured were consistent with the reported solubility of amorphous silica at the internal solution temperature in the middle of Cell 2 (Figure 2 and Table 2), except for samples collected at temperatures of 64.5, 76.7, 111.7, and 152.6 °C. The average difference between the measured temperature and the temperature calculated from the silica solubility, except for inconsistent data, was 9.8 °C (Table 2). The error may be due to temperature fluctuation in the cell or the reprecipitation of silica after Cell 2, particularly during sampling. However, the temperatures estimated for the reaction occurring in Cell 2 were sufficiently reasonable to conduct hydrothermal Cr spinel formation experiments under low-temperature conditions within ± 9.8 °C of the estimated temperature error.

Dissolved silica concentrations of samples collected at temperatures of 64.5, 76.7, 111.7, and 152.6 °C were much lower than the amorphous silica solubility at these temperatures, which appeared to follow the solubility of quartz. XRD analysis of the recovered solid samples indicated only the presence of amorphous silica, which suggested that transformation from amorphous silica to quartz did not occur in reaction Cell 2. Therefore, these results were likely explained by rapid decompression and boiling of the solutions during sampling. Boiling would have caused the precipitation of silica in the sampling line, although we could not examine whether the precipitated silica phase was quartz. The results suggested that careful sampling is required for the flow-through experiments.

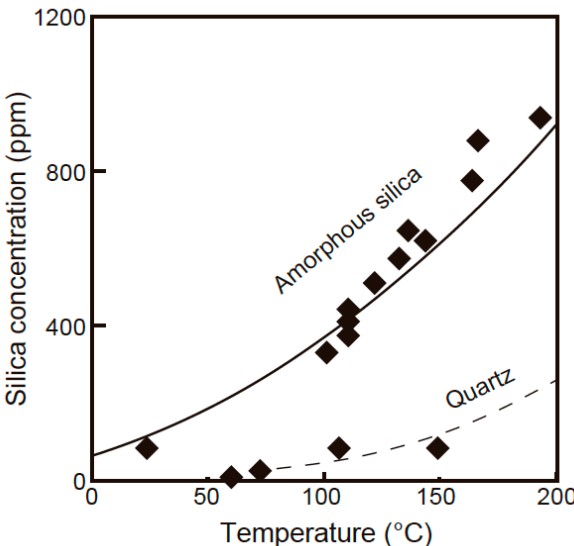

**Figure 2.** Dissolved silica concentrations of solution samples (black diamond) plotted as a function of temperature measured by a thermocouple in Cell 1 for temperature calibration with the amorphous silica (solid line) and quartz (dashed line) solubility curves reported in [17]. Amorphous silica and quartz solubility curves were originally described in [19–22].

**Table 2.** Difference between the measured temperature and temperature calculated from the silica solubility [17].

| Measured Temperature | Silica Concentration | Temperature Calculated from the Silica Solubility | Difference of Measured and Calculated Temperature |
|---|---|---|---|
| (°C) | (ppm) | (°C) | (°C) |
| 28.2 | 79.1 | 5.7 | 22.5 |
| 64.5 | 1.5 | −104.9 | |
| 76.7 | 19 | −47.6 | |
| 105.2 | 325.5 | 91.0 | 14.2 |
| 111.7 | 75.1 | 3.3 | |
| 113.1 | 375 | 102.5 | 10.6 |
| 114.2 | 437.1 | 115.8 | 1.6 |
| 114.2 | 409.3 | 110.0 | 4.2 |
| 114.9 | 370.7 | 101.5 | 13.4 |
| 124.6 | 508.8 | 129.9 | 5.3 |
| 136.0 | 570.7 | 141.3 | 5.3 |
| 139.4 | 642.9 | 153.9 | 14.5 |
| 146.5 | 617.1 | 149.5 | 3.0 |
| 152.6 | 77.9 | 5.0 | |
| 167.5 | 776.4 | 175.3 | 7.8 |
| 169.9 | 877.1 | 190.4 | 20.5 |
| 195.7 | 944.4 | 200.0 | 4.3 |
| | | | Average: 9.8 |

### 3.2. Solution Chemistry in Fe-Cr Hydroxide Alteration Experiments

The pH of the solution samples in runs 1–4 gradually increased (runs 2 and 4) or was constant (runs 1 and 3) after the first few to several hours of the experiment without outliers (Figure 3a and Table 3), which indicated that the effect of sampling on the water chemistry was likely minimal. The rapid increase (runs 2 and 3) and decrease (runs 1 and 4) of pH at the beginning of the runs indicated unstable solution chemistry in the flow-through system, probably due to heterogeneous reactions with the $Fe^{2+}_{(aq)}$ supply or hydrothermal alteration of the Fe-Cr hydroxides with penetration of the acidic solution to pore spaces in

the cells right after the start of the experiments. The pH was kept constant or gradually increased to a value from 2.53 to 4.69, which suggested the solution compositions were relatively homogeneous throughout the duration of the experiments. The Fe concentrations of the solution samples in runs 1 and 2 were <0.73 ppm (Figure 3b and Table 3), which indicates most of the $Fe^{2+}_{(aq)}$ supplied from Cell 1 was consumed for Fe-Cr hydroxide alteration in Cell 2. The Fe concentration of the solution sample in run 3 rapidly increased to 14.4 ppm after 20 h of the experiment and then remained constant at ca. 10 ppm. The Fe concentration in run 4 was the highest at 101 ppm after 1 h, rapidly decreased to less than the detection limit after 66 h, and then slightly increased and decreased from 0.28 to 0.89 ppm at the end of the experiment. The flow rate of runs 1 and 2 was set to 2.0 $cm^3$/min, whereas the flow rate of runs 3 and 4 was set to 1.0 $cm^3$/min (Table 2). These results suggested that the $Fe^{2+}_{(aq)}$ supply from Cell 1 likely increased with a decrease of the flow rate and could not all be consumed for the Fe-Cr hydroxide alteration in runs 3 and 4. The Fe concentration at the beginning of run 4 was the highest (101 ppm at 1 h), which may be due to a high dissolution rate of magnetite at higher temperature (200 °C) and a low solid-liquid ratio (low amount of magnetite in Cell 1 for run 4, Table 1) in Cell 1 than other runs (150 and 170 °C; Table 1) in addition to a slower flow rate.

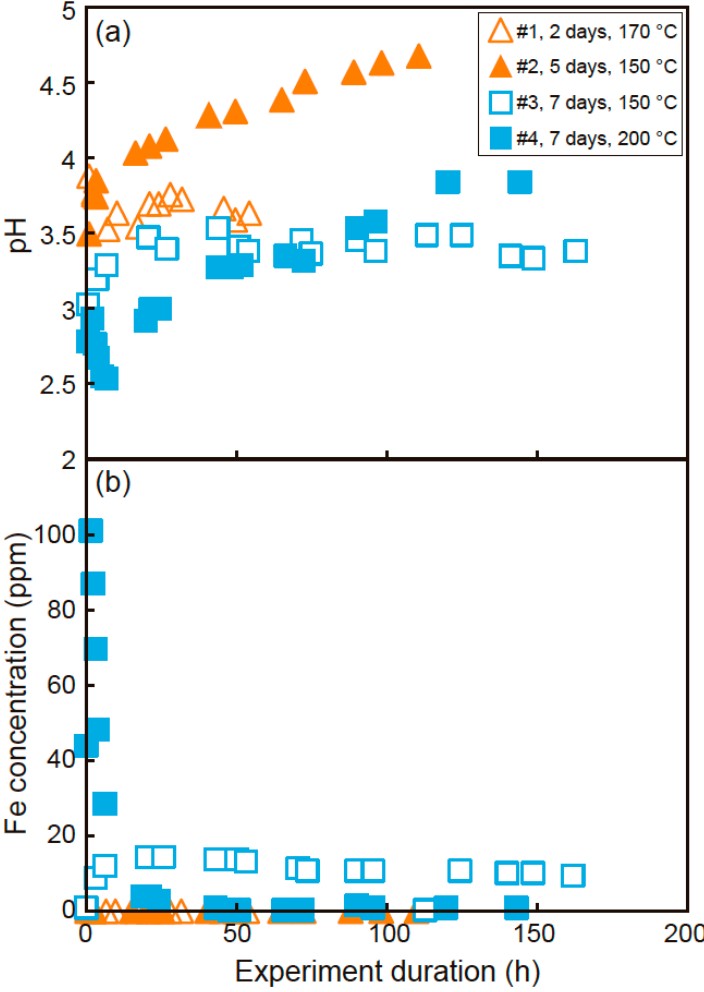

**Figure 3.** (**a**) pH and (**b**) Fe concentration for solution samples as a function of experiment duration in runs 1–4.

**Table 3.** pH and Fe concentrations of solution samples from runs 1–4 at various times during the experiments.

| Run No. | Experimental Duration | pH | Fe Concentration (ppm) |
|---|---|---|---|
| | (hour) | | |
| 1 | 0 | 3.88 | 0.73 |
| | 1 | 3.76 | 0.72 |
| | 2 | 3.74 | <d.l. |
| | 6 | 3.52 | <d.l. |
| | 9 | 3.63 | <d.l. |
| | 16 | 3.54 | <d.l. |
| | 20 | 3.7 | <d.l. |
| | 23 | 3.69 | <d.l. |
| | 27 | 3.76 | <d.l. |
| | 31 | 3.73 | <d.l. |
| | 45 | 3.66 | <d.l. |
| | 49 | 3.59 | <d.l. |
| | 53 | 3.64 | <d.l. |
| 2 | 0 | 3.50 | 0.07 |
| | 1 | 3.77 | 0.09 |
| | 2 | 3.85 | 0.05 |
| | 15 | 4.04 | 0.05 |
| | 20 | 4.08 | 0.04 |
| | 25 | 4.13 | 0.05 |
| | 40 | 4.28 | 0.05 |
| | 49 | 4.32 | 0.05 |
| | 64 | 4.39 | 0.05 |
| | 72 | 4.51 | 0.05 |
| | 88 | 4.58 | 0.04 |
| | 98 | 4.64 | 0.04 |
| | 110 | 4.69 | 0.04 |
| 3 | 0 | 3.02 | 0.38 |
| | 3 | 3.19 | 8.36 |
| | 6 | 3.28 | 11.9 |
| | 20 | 3.46 | 14.4 |
| | 26 | 3.39 | 14.3 |
| | 43 | 3.52 | 13.6 |
| | 50 | 3.41 | 13.3 |
| | 53 | 3.37 | 13.0 |
| | 71 | 3.45 | 11.1 |
| | 74 | 3.36 | 10.7 |
| | 90 | 3.45 | 10.4 |
| | 96 | 3.37 | 10.7 |
| | 113 | 3.48 | failed |
| | 125 | 3.48 | 10.3 |
| | 141 | 3.34 | 9.95 |
| | 149 | 3.33 | 9.61 |
| | 163 | 3.37 | 9.20 |
| 4 | 0 | 2.77 | 43.5 |
| | 1 | 2.92 | 101 |
| | 2 | 2.75 | 86.6 |
| | 3 | 2.66 | 69.5 |
| | 4 | 2.54 | 48.2 |
| | 6 | 2.53 | 28.2 |
| | 19 | 2.91 | 3.84 |
| | 21 | 2.99 | 3.37 |
| | 24 | 2.98 | 2.67 |
| | 43 | 3.26 | 0.38 |
| | 48 | 3.26 | 0.02 |
| | 51 | 3.28 | 0.02 |
| | 66 | 3.34 | 0.00 |
| | 72 | 3.31 | 0.00 |
| | 90 | 3.53 | 0.89 |
| | 96 | 3.57 | 0.84 |
| | 120 | 3.83 | 0.55 |
| | 144 | 3.83 | 0.28 |

d.l.: detection limit.

### 3.3. Hydrothermal Chromian Spinel Formation from Fe-Cr Hydroxides

The Fe-Cr hydroxide starting material only showed broad peaks attributed to the 2-line ferrihydrite-like structure in the XRD profile (Figure 4a). The XRD profiles of solid samples from runs 1–3 showed peaks attributed to goethite, hematite, and a spinel-group mineral (Figure 4b–d), which indicated a spinel-group mineral was formed at 150 and 170 °C. The profile for the solid sample of run 4 showed only peaks attributed to hematite (Figure 4e), which suggested the lower solid-liquid ratio and temperature selectively promoted transformation from Fe-Cr hydroxides to hematite. Mössbauer spectroscopy also indicated that ferrihydrite, goethite, and hematite were present in the solid sample from run 3 (Figure 5). The peaks of a spinel-group mineral [23,24] could not be clearly observed, probably due to the small quantity. The full width at a half-maximum value for the peaks attributed to ferrihydrite indicated 0.49 mm/s, which was much larger than the theoretical value for ferrihydrite (0.19 mm/s)-like natural low-crystalline ferrihydrite with impurities [25], suggesting that the peaks for spinel-group mineral may be covered by the peaks for ferrihydrite. Mössbauer spectroscopy also indicated that Fe-Cr hydroxides accounted for most of the recovered samples so that most of the starting materials remained after the experiments.

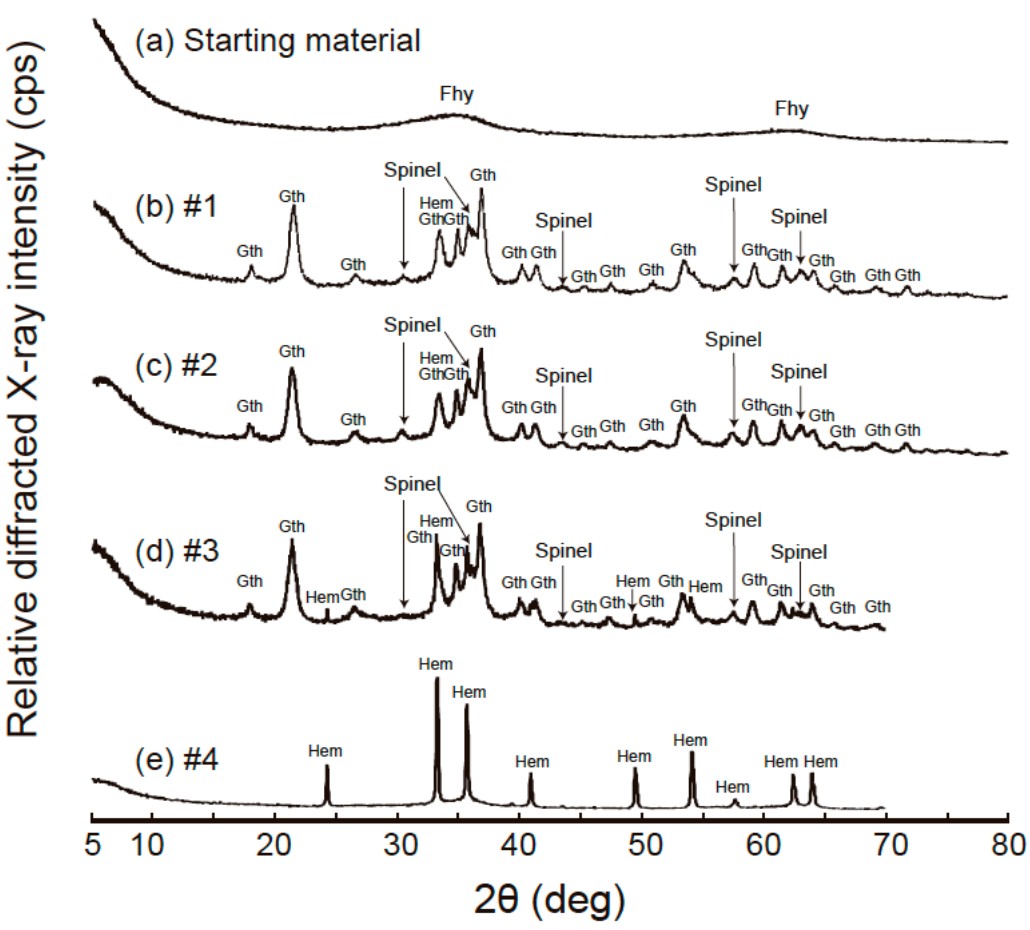

**Figure 4.** XRD profiles of (**a**) the starting material and solid samples from runs (**b**) 1, (**c**) 2, (**d**) 3, and (**e**) 4 recovered from Cell 2 after the experiments. Fhy: Ferrihydrite, Gth: Goethite, Hem: Hematite.

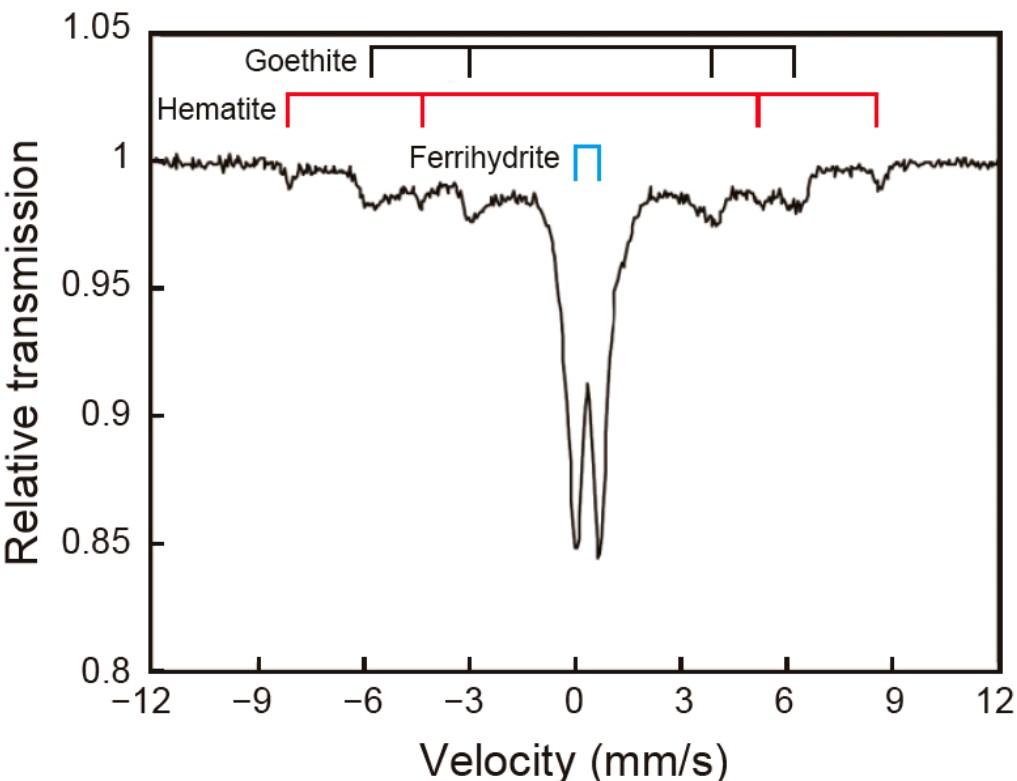

**Figure 5.** Mössbauer spectrum of the solid sample recovered from run 3 with peaks attributed to goethite (black line), hematite (red line), and ferrihydrite (blue line).

FE-SEM observations of the solid sample recovered from run 3 revealed aggregates of euhedral crystals on rounded grains of Fe-Cr hydroxide (Figure 6a, sample from run 3). Such aggregates were a mixture of octahedral and hexagonal platy crystals surrounded by <500 nm needle-like crystals (Figure 6b). The needle-like crystals were interpreted as goethite; the morphology was dominated by {100}, which is the typical crystal facet for goethite. The goethite aggregates accounted for most of the crystal phases in the FE-SEM observations, which was consistent with the XRD analysis (Figure 4b–d). The hexagonal platy crystals had a similar hexagonal morphology to hematite with dominant facets of {001} and {102} (Figure 6b, upper right). EDS analysis combined with FE-SEM observations indicated the Cr/Fe ratio in the goethite crystal was almost identical to that of the starting material at a Cr/Fe ratio of 0.2 to 0.25. The octahedral crystals had similar morphology to that of a spinel-group mineral revealing {111} facets (Figure 6b, lower left). The crystals contained Cr with Cr/Fe ratios of 0.03 to 0.1, which was lower than that of goethite or the starting material. TEM observations showed ca. 400 nm aggregates of ca. 50 nm diameter particles (Figure 6c) with an identical size to the octahedral crystals in the FE-SEM observation (Figure 6b) and spinel-like morphology. Diffraction patterns obtained from the entire area of Figure 6c were attributed to (311), (220), and (111) crystal facets of magnetite and chromite (Figure 6d), which indicated the formation of spinel in the flow-through experiment. The particles comprising the aggregates showed similar morphology to goethite; however, the crystal phases of the particles could not be identified, because the diffraction peaks of goethite were weak or partially overlapped with those of magnetite or chromite. TEM-EDS analysis indicated that the aggregates contained Cr and Fe at a Cr/Fe ratio of 0.25. Hence, the synthesized chromian spinel was classified as Cr-rich magnetite according to the nomenclature and classification by the International Mineralogical Association in [26] due to the dominance of Fe in the stoichiometric ratios of the oxyspinel group. Note that maghemite, $Fe^{3+}$-bearing oxyspinel group, is difficult to be formed under $Fe^{2+}_{(aq)}$-supplied conditions, which was discussed in [12].

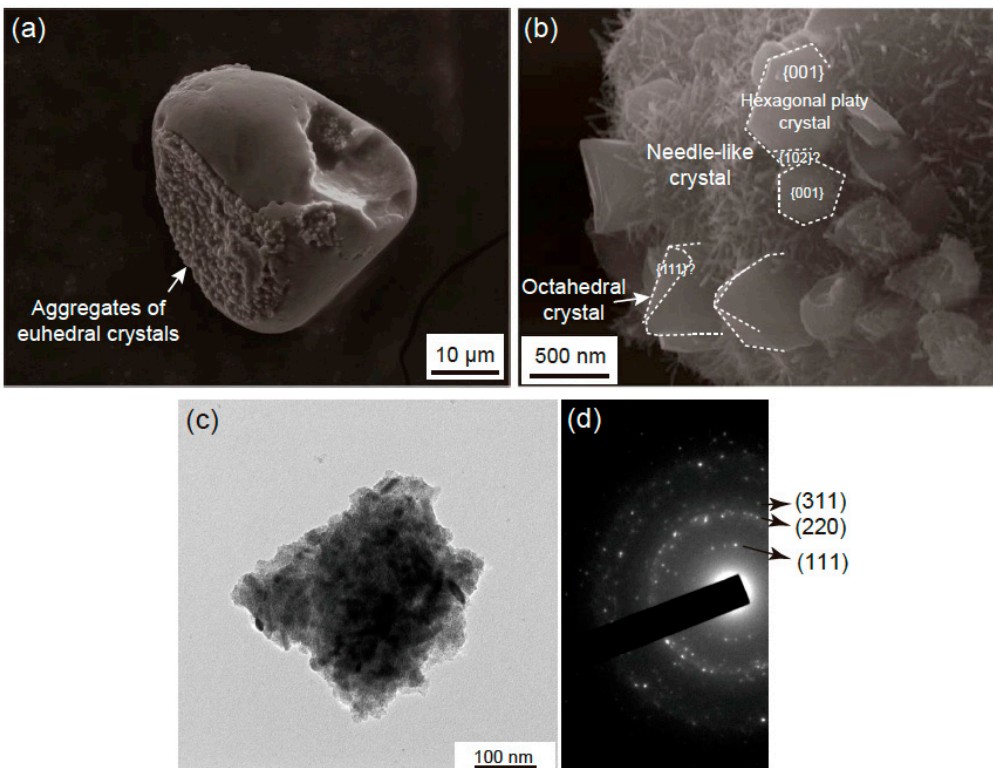

**Figure 6.** (**a**,**b**) FE-SEM images of samples recovered from run 3. (**a**) Several euhedral crystals formed aggregates on the rounded particle starting materials. (**b**) Needle-, hexagonal platy, and octahedral crystals were included in the aggregations. (**c**) TEM image of sample recovered from run 3 with ca. 400 nm aggregate of ca. 50 nm diameter particles. The aggregate showed spinel-like morphology. (**d**) Electron diffraction pattern corresponding to the aggregate shown in (**c**). The major rings in the pattern were attributed to (311), (220) and (111) crystal facets of magnetite and chromite.

Compositions of the solution samples of runs 1–4 were plotted with the stability field of magnetite at 150, 170, and 200 °C (Figure 7). The $Fe^{2+}_{(aq)}$ activities and pH of the solutions at 150, 170, and 200 °C were calculated using the SpecE8 and React module of the Geochemist Workbench (GWB) 2021 (Aqueous Solutions LLC, Champaign, IL, USA). Note that the plotted $Fe^{2+}_{(aq)}$ activities of the solutions indicated the minimum values of the flow-through system because $Fe^{2+}_{(aq)}$ could be consumed in Cell 2, which resulted in low Fe concentrations of the solution samples. In run 2, the solution maintained a low Fe concentration with a pH increase from 3.53 to 4.71, which approached the stability field of magnetite, shown on the right side of the diagram (Figure 7), while the solution of run 3 showed a higher Fe concentration than that in run 2 and kept a narrow range of pH from 3.05 to 3.55 with an approach to the stability field of magnetite, as seen in the upper side of the diagram. These results suggested that some reactions involving Fe-Cr hydroxides caused a pH increase under low $Fe^{2+}_{(aq)}$ activity. In the experimental system, Fe-Cr hydroxides could be dissolved in the acidic solution in Cell 2, as in the following reaction at a Cr/Fe ratio of 0.25:

$$(Fe_{0.8}, Cr_{0.2})(OH)_2^+ + 2H^+ \rightarrow 0.8Fe^{3+} + 0.2Cr^{3+} + 2H_2O \tag{3}$$

Note that $(Fe_{0.8}, Cr_{0.2})(OH)_2^+$ was the dominant dissolved species for Fe under the experimental conditions. This reaction consumed $H^+$, which might have caused the pH increase in run 2. Cr-rich magnetite formation by Fe-Cr hydroxide alteration with $Fe^{2+}_{(aq)}$

supply could be described by the following reaction when all Cr in Fe-Cr hydroxides are converted into magnetite:

$$2(Fe_{0.8}, Cr_{0.2})(OH)_2{}^+ + Fe^{2+}{}_{(aq)} \rightarrow Fe\ (Fe_{1.6}, Cr_{0.4})O_4 + 4H^+ \tag{4}$$

Reaction (4) caused a decrease of pH in the solutions. Solid samples recovered from runs 1–3 also contained goethite and hematite, suggesting that $Fe^{2+}$-bearing Fe-Cr hydroxides were simultaneously dehydrated, which could be described by the following reactions modified from [27]:

$$10\{Fe(OH)_n(H_2O)_{(6-n)}\}^{(3-n)+} \rightarrow 5Fe_2O_3 \cdot 9H_2O + (30 - 10n)H_3O^+ + (6 + 10n)H_2O \tag{5}$$

$$10\{Fe(OH)_n(H_2O)_{(6-n)}\}^{(3-n)+} \rightarrow 10FeOOH \cdot 4H_2O + (30 - 10n)H_3O^+ + (6 + 10n)H_2O \tag{6}$$

These reactions decreased the pH with a $Fe^{2+}$ addition to Fe-Cr hydroxides at $n < 3$ and no pH change occurred without a $Fe^{2+}$ addition at $n = 3$. Solution chemistry and the constituent mineral assemblages of the samples from runs 1–3 suggested that Cr-rich magnetite was formed under the balance of reaction (3)'s dissolution of Fe-Cr hydroxides, (4)'s Cr-rich magnetite formation with $Fe^{2+}$ addition, and (5)'s and (6)'s dehydration of Fe-Cr hydroxides. The balance was likely maintained in runs 1 and 3, which resulted in a relatively constant pH, whereas the gradual pH increases in the solution samples of runs 2 and 4 were likely governed by Fe, $Cr(OH)_2{}^+$ dissolution under the experimental conditions.

Synthesized Cr-rich magnetite consisted of nanosized columnar particles (Figure 6c), which suggested that Cr-rich magnetite is formed from other iron minerals, possibly goethite. As shown in reaction (1), hematite could be redox-independently transformed to magnetite with a $Fe^{2+}{}_{(aq)}$ supply via dissolution or reprecipitation at low-temperature-reducing hydrothermal conditions (150 °C) where redox kinetics was sluggish [12]. If it was assumed that a similar transformation could occur in the present experiments, then goethite could also be the intermediary according to the following reaction:

$$2Fe_nCr_{(1-n)}OOH + Fe^{2+}{}_{(aq)} \rightarrow 2H^+ + Fe_{2n+1}Cr_{(2-2n)}O_4\ (0 < n < 1) \tag{7}$$

Goethite was abundant in the recovered solid samples and showed an almost identical Cr/Fe ratio (0.2 to 0.25) with that of the starting material, which supported that the main intermediary for the formation of hydrothermal Cr-rich magnetite was goethite (reaction (7)). When considering the lower Cr/Fe ratios of synthesized Cr-rich magnetite (0.03 to 0.25 from FE-SEM-EDS and TEM-EDS analyses), the transformation from goethite to Cr-rich magnetite involved the redistribution of Cr during dissolution or reprecipitation. Cr may be incorporated into the goethite rather than magnetite, possibly because of the higher capability of goethite to incorporate Cr in the crystal structure.

Previous studies have reported that the transformation of poorly ordered ferrihydrite into crystalline goethite occurs rapidly in the presence of ferrous iron, which acts as a catalyst [28,29], and that Cr stabilizes goethite against dissolution by acidic solutions [30]. These observations suggest that a $Fe^{2+}{}_{(aq)}$ supply and the presence of Cr in the present experimental system stabilized goethite and suppressed Cr-rich magnetite formation. Accordingly, the relatively high Fe concentration and constant pH in run 3 could also be interpreted as a result of the rapid transformation of Fe-Cr hydroxides into goethite with a small amount of Cr-rich magnetite formation at a slower flow rate (1 cm$^3$/min). If this hypothesis is correct, then the dissolution of Fe-Cr hydroxides would be inhibited at the slower flow rate, and $Fe^{2+}{}_{(aq)}$ would be efficiently adsorbed from the acid solution into the Fe-Cr hydroxides with a sufficient retention time. Goethite formed by the dehydration of Fe-Cr hydroxides was relatively stable due to the $Fe^{2+}{}_{(aq)}$ supply and the presence of Cr. Under such conditions, much of the $Fe^{2+}{}_{(aq)}$ would pass through Cell 2 without reacting with the Fe-Cr hydroxides, which would result in a relatively high Fe concentration and constant pH, as observed in run 3.

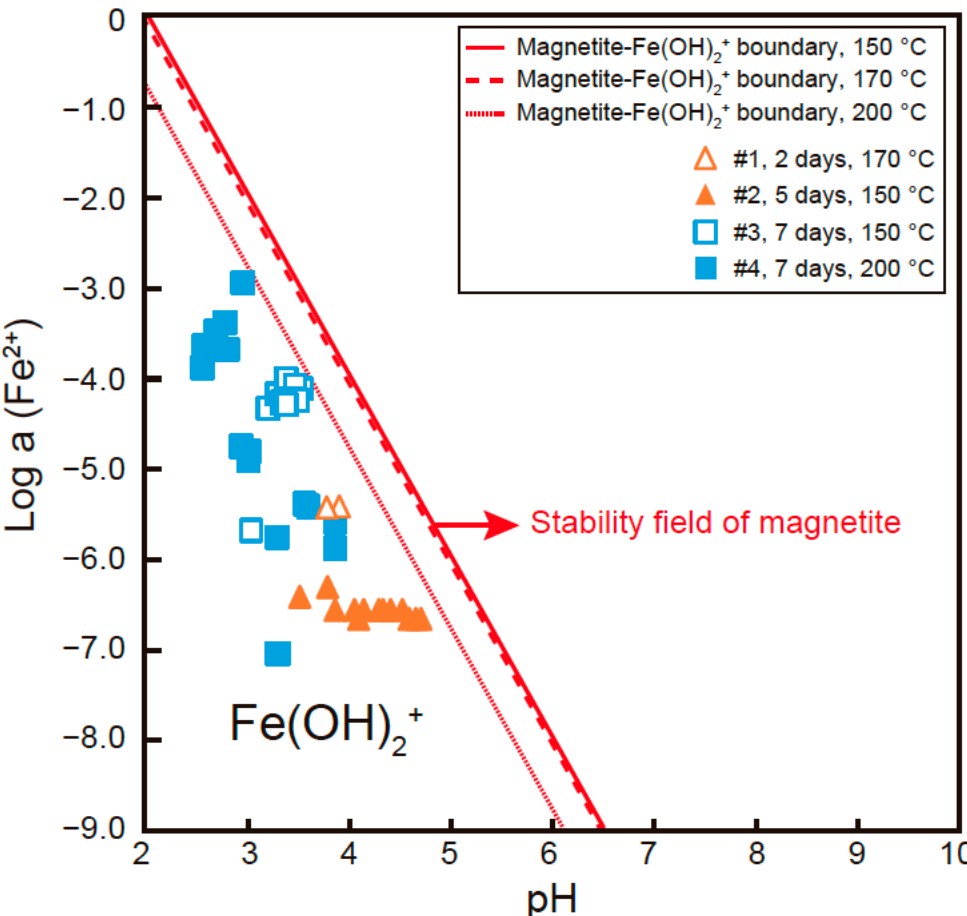

**Figure 7.** pH-log activity stability fields of possible Fe phases at 150, 170, and 200 °C. Calculated solution compositions of the samples from runs 1–4 at 150, 170, and 200 °C are plotted in the phase diagram.

Although ferrihydrite was not in the stable iron phase under the experimental temperatures, the Mössbauer spectroscopy results indicated that most Fe-Cr hydroxides remained after the experiments. This result suggested that the existence of Cr or other experimental conditions improved the stability of Fe-Cr hydroxides. Stability of Fe-Cr hydroxides in acidic oxygen-free solution at 150–200 °C has not been reported in previous studies, however, it may be predicted from the solubility and structural features at ambient temperature. Papassiopi et al. (2014) [31] have reported that mixed $Fe_{0.75}Cr_{0.25}(OH)_3$ hydroxides have a similar structure to pure ferrihydrite, and the solubility could be predicted as a solid solution of two hydroxides. Mixed $Fe_{0.75}Cr_{0.25}(OH)_3$ indicated lower solubility than pure $Cr(OH)_3$ in the pH range of the experiment at ambient temperature [31], indicating that the solubility of Fe-Cr hydroxides is higher than that of pure ferrihydrite and a Cr addition may increase the solubility of ferrihydrite. When considering the structural features of $Cr^{3+}(OH)_3$, Amonette and Rai (1990) [16] suggested that the existence of $Cr(OH)_3$ was responsible for the persistence of the non-crystalline state and the inhibition of the conversion to a crystalline phase due to the relative inert coordination sphere of $Cr^{3+}$ to ligand-exchange reactions. Campbell et al. (2002) [32] reported that the presence of Si also significantly increased the temperature for the transformation from ferrihydrite to hematite due to the suppression of crystal growth along [hk0]. Considering these previous reports, our result suggested that a Cr addition could increase the stability of ferrihydrite by altering structural features. The result of run 4 indicated that most Fe-Cr hydroxides transformed to hematite at 200 °C (Table 1 and Figure 4e), suggesting that the stabilization is ineffective at >200 °C.

## 4. Conclusions

Hydrothermal chromian spinel synthesis via non-redox transformation was conducted using a flow-through reactor to examine the physicochemical conditions at 150, 170, and 200 °C and 5 MPa under mildly acidic oxygen-free conditions (pH > 3–5). XRD data of the recovered solid samples from the reaction in Cell 2, where Fe-Cr hydroxide was altered with $Fe^{2+}_{(aq)}$ supplied by acid-leaching of magnetite, indicated that a chromian spinel was formed from Fe-Cr hydroxide at 150 and 170 °C together with goethite and hematite. The synthesized chromian spinel was classified as Cr-rich magnetite by the chemical composition. FE-SEM and TEM observations suggested that it was most likely the goethite aggregates that transformed into Cr-rich magnetite with redistribution of Cr via dissolution and reprecipitation. Although chromian spinel has been recognized as a high-temperature igneous mineral precipitated from magma in previous studies, the present results indicate that it could also be formed in mild to low-temperature hydrothermal systems (ca. <200 °C).

**Author Contributions:** Conceptualization, T.O.; methodology, T.O., J.Y., M.N. and Y.O.; formal analysis, J.Y., M.N. and Y.O.; investigation, J.Y., M.N. and Y.O.; resources, D.K. and Y.K.; writing—original draft preparation, Y.O.; writing—review and editing, J.Y., M.N., D.K., Y.K., T.O. and T.S.; visualization, Y.O.; supervision, T.O. and T.S.; project administration, T.O.; funding acquisition, T.O. All authors have read and agreed to the published version of the manuscript.

**Funding:** This work was conducted by Kakenhi Grants-in-Aid (No. 26709069, No. 17K19081 and No. 19K15486) given to T.O from the Japan Society for the Promotion of Science (JSPS).

**Data Availability Statement:** Not applicable.

**Acknowledgments:** The authors thank A. Kurishiba and other staff members in the laboratory of Nano-Micro Material Analysis, Joint-use Facilities, Faculty of Engineering, Hokkaido University for support with FE-SEM and TEM observations; T. Yamazaki at the Institute of Low-Temperature Science, Hokkaido University for support with TEM observations; and F. Chikanda for support with English correction. This work was conducted at the Joint-use Facilities: Laboratory of Nano-Micro Material Analysis, Hokkaido University, and supported by the Nanotechnology Platform program of the Ministry of Education, Culture, Sports, Science and Technology (MEXT), Japan. Y.J. contributed to this work as a private venture, which was not in the capacity of an employee of the Jet Propulsion Laboratory, California Institute of Technology.

**Conflicts of Interest:** The funders had no role in the design of the study; in the collection, analysis, or interpretation of data; in the writing of the manuscript; or in the decision to publish the results.

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
