# Peer review of "Low-Temperature Hydrothermal Synthesis of Chromian Spinel from Fe-Cr Hydroxides Using a Flow-Through Reactor"

_minerals, doi:10.3390/min12091110_

Round 1

Reviewer 1 Report

In this paper, the conditions for the hydrothermal synthesis of chromium spinel and the corresponding chromium redistribution were investigated in a flow-through reactor under anaerobic and moderately acidic conditions. However, a proper explanation is missing from the manuscript. The complete manuscript must be rewritten in technically appropriate content and grammatically correct English. The purpose of the manuscript is not clear. I feel the manuscript cannot be accepted in the current format, so it needs to be revised in good faith by processing the comments given below.

1.      The language logic of the introduction is not clear enough and the purpose is not clear enough.

2.      The clarity of the picture needs to be enhanced.

3.      On page 9, parts 242-264 are confused in tense, leading to ambiguity.

4.      Page 2, line 76. How is 0.2 to 0.33 determined? Are relevant experiments performed for each segment in between, or is this range randomly selected? Please explain

5.      Where did the Na+ introduced in the experiment go? How to remove?

6.      The author mentioned that it is necessary to finally synthesize part of chromium spinel and part of iron ore, and how to separate or remove impurities in the future. Can the synthesized sample continue to be used or can spinel be synthesized just for testing?

7.      The author has mentioned that the test of the synthesized sample shows that it contains iron-chromium hydroxide. What is the reason? How to solve it? Just citing other people's literature is not enough.

Author Response

Dear Reviewer 1,

We greatly appreciate your valuable comments. According to your comments our manuscript was revised as follows.

  1. The language logic of the introduction is not clear enough and the purpose is not clear enough.
  2. The author mentioned that it is necessary to finally synthesize part of chromium spinel and part of iron ore, and how to separate or remove impurities in the future. Can the synthesized sample continue to be used or can spinel be synthesized just for testing?

Response:
These comments are related to the ambiguity of objective in introduction. Our paper conducted the artificial hydrothermal crystallization of spinel; however, the objective is to verify hydrothermal chromian spinel formation in natural hydrothermal environments. To clarify the objective and the significance of our work, we modified paragraph 1 and combined it with paragraph 2. Sentences related to engineering problems were omitted because it made the readers confused. Paragraph 3 and First sentence of the abstract were also omitted.

Revised introduction

Chromian spinel has been recognized as a typical high-temperature igneous mineral recording magmatic process in nature [1–6]. Recent petrological investigations have suggested that chromian spinel could be nucleated from hydrothermal fluids during metamorphic and/or metasomatic events at lower temperature [7–9]. On the other hand, it also has been interpreted as relics of igneous phases [10]. Existence of hydrothermal chromian spinel formation in nature is therefore still controversial, while it would affect previous interpretation of the igneous rocks. Experimental studies can verify the feasibility of hydrothermal chromian spinel in nature. Chromian spinel formation under hydrothermal conditions (ca. <300 °C) has been performed using a batch-type pressure vessel in a pioneering experiment of engineering field [11]; however, such closed system equipment cannot demonstrate chromian spinel formation in a natural oxygen-free hydrothermal system with fluid-circulation. When assuming the formation in natural hydrothermal system, most possible precursors for chromian spinel are Fe-Cr hydroxides because Fe2+(aq) in hydrothermal fluid is oxidized when mixed with dissolved Cr6+ by the difference of the redox potential, resulting in precipitation of Fe-Cr hydroxides. Then reduction of Fe-Cr hydroxides is required to form chromian spinel in classic thermodynamical calculation at equilibrium. However, a previous experimental study has indicated that the non-redox transformation of Fe-bearing minerals could occur under H2-rich acidic hydrothermal conditions via dissolution/reprecipitation as following this reaction [12]:

Fe2O3 + Fe2+(aq) + H2O = Fe3O4 + 2H+      (1)

The reaction (1) is reversible and redox-independent replacement of magnetite by hematite has been experimentally investigated and widely observed in natural iron ores in recent studies [13–15]. The finding led us to the idea that chromian spinel can also be formed by similar non-redox acid-base reaction at low-temperature hydrothermal conditions. In this study, we have developed a high-pressure flow-through reactor to conduct hydrothermal chromian spinel synthesis from Fe-Cr hydroxides with an aqueous ferrous iron (Fe2+(aq)) supply under mildly acidic conditions (pH of 3–5) with the assumption that the conditions of oxygen-free hydrothermal environments. The flow-through experiment was performed at various solid-liquid ratios, initial pHs of the supplied solutions and flow rates to examine the physicochemical conditions, especially lower limit of temperature feasible for chromian spinel formation at 150, 170 and 200 °C.

  1. The clarity of the picture needs to be enhanced.

Response:

According to the comment, the clarity of the pictures was improved.

  1. On page 9, parts 242-264 are confused in tense, leading to ambiguity.

Response:

According to the comment, we corrected the tense in the paragraph.

  1. Page 2, line 76. How is 0.2 to 0.33 determined? Are relevant experiments performed for each segment in between, or is this range randomly selected? Please explain

Response:

According to the comment, we added following sentences in Material and Methods.

Fe-Cr hydroxides were basically synthesized at the Cr/Fe ratio of 0.25 by the method in [16]. Synthesis of homogeneous Cr-rich starting material was difficult due to the rapid precipitation of Cr hydroxides under atmospheric condition; therefore, we varied the Cr/Fe ratios in the starting materials in the narrow range from 0.2–0.33.

  1. Where did the Na+ introduced in the experiment go? How to remove?

Response:

We did not remove Na+ in the solution, so that Na+ is dissolved in acid solutions. Our objective is not to synthesize pure chromian spinel but to demonstrate chromian spinel formation in natural hydrothermal conditions and we did not need to remove Na+ because it is a ubiquitous dissolved element. Na+ was rather added to the solution to set the ionic strength to ca. 0.1 in the system.

  1. The author has mentioned that the test of the synthesized sample shows that it contains iron-chromium hydroxide. What is the reason? How to solve it? Just citing other people's literature is not enough.

Response:

According to the comment, we revised the last paragraph as follows.

Revised paragraph

Although ferrihydrite is not the stable iron phase under the experimental temperatures, the Mössbauser spectroscopy results indicate that most Fe-Cr hydroxides remained after the experiments. The result suggested the existence of Cr or other experimental conditions improve the stability of Fe-Cr hydroxides. Stability of Fe-Cr hydroxides in acidic oxygen-free solution at 150–200 °C has not been reported in previous studies, however, it may be predicted from the solubility and structural features at ambient temperature. Papassiopi et al. (2014) [29] have reported that mixed Fe0.75Cr0.25(OH)3hydroxides have a similar structure to pure ferrihydrite, and the solubility could be predicted as a solid solution of two hydroxides. Mixed Fe0.75Cr0.25(OH)3 indicated lower solubility than pure Cr(OH)3 in the pH range of the experiment at ambient temperature [29], evoking that solubility of Fe-Cr hydroxides are higher than pure ferrihydrite and Cr addition may increase the solubility of ferrihydrite. When considering structural features of Cr3+(OH)3, Amonette and Rai (1990) [16] suggested that existence of Cr(OH)3 is responsible for the persistence of the non-crystalline state and the inhibition of the con-version to a crystalline phase due to the relative inert coordination sphere of Cr3+ to ligand-exchange reactions. Campbell et al. (2002) [30] reported that the presence of Si also significantly increases the temperature for the transformation from ferrihydrite to hematite due to the suppression of crystal growth along [hk0]. Considering with these previous reports, our result suggests that the Cr addition could increase the stability of ferrihydrite by altering structural features. The result of run #4 indicated that most Fe-Cr hydroxides transformed to hematite at 200 °C (Table 1 and Figure 4e), suggesting that the stabilization is ineffective at > 200 °C.

We also found some mistakes in the manuscript and corrected as follows.
1) Daisuke Kawamoto’s address was corrected (Ridaimachi → Ridaicho)
2) Solution sampling was conducted from back pressure valve.

3) Fe2+ in reaction formula was replaced with Fe2+(aq).
4) References related to silica solubility were corrected (Morrey 1951→1962) and added (Fournier and Rowe, 1977) in the list.

Reviewer 2 Report

I read the manuscript of Ohtomo et al. (minerals-1856085) with much interest as it deals with a significant and potentially interdisciplinary topic, which is the synthesis of spinel. The paper is very well written, it involves very good methods for material characterisation and the interpretations are in general well addressed.

I have a few concerns and recommendations for the authors:

1.       The paper deals with the artificial hydrothermal crystallisation of spinel and therefore paragraphs 1 and 3 of the Introduction (although correct) are out of context. I strongly suggest omitting them.

2.       After the above, the Introduction will be (and still is) a bit small and a deeper presentation of the problem statement is very much desirable. Also, it was unclear to me what the significance of this study is. Do you intend to address and contribute to the solution of engineering problems related to corrosion or something else? Please clarify and highlight the significance of the research.

3.       In Table 1, a brief description of how the experimental conditions were selected would be highly desirable. A minor point is that in my opinion the table needs some rectifications and to tidy up the headers.

4.       Chapter 3.3 is very important but I have the impression that more citations are required to substantiate the interpretations. For example, an important issue is that the formation of spinel after goethite requires reducing conditions. Any comment about that? Is there any major change in terms of dissolved oxygen (or oxygen fugacity?) during the experiments? The recommended reactions are acceptable but is there any further literature supporting them? It would be good to cite more references on the formation of spinels after oxides. Reactions 6 and 7 are not very well supported. Also, the text in lines 336-348 is a bit rushed and requires further elaboration. The statement in lines 344-345 is arbitrary and requires better substantiation.

5.       Finally, after revising the introduction, the conclusions should be revised in a way to respond to the problem statement apart from the major findings.

Some minor points in chapter 3.3:

I suggest replacing the term “granular” (describing the crystal habit of the synthesised spinel) with “octahedral”. In my view, Fig. 6B (lower left) shows clearly the top part of an octahedron!

Please replace the term “plate-like” with “platy”, which is the usual term to describe hematite and other crystals with this crystal habit. Moreover, “hexagonal platy” would be even better.

Please replace {001} with {100} in line 246, which is the typical (and what you show us from your observations) family of faces for this columnar goethite.

Please replace {001} with {111} in line 253, which is the typical family of faces of spinel (octahedral).

Also in my view, Figure 7 is not a phase diagram sensu stricto and I suggest changing this term when you cite it in your text.

I would be very happy to see this paper published in Minerals

Best regards

Author Response

Dear reviewer 2,

We greatly appreciate your valuable comments. According to your comments our manuscript was revised as follows.

1. The paper deals with the artificial hydrothermal crystallisation of spinel and therefore paragraphs 1 and 3 of the Introduction (although correct) are out of context. I strongly suggest omitting them.

  1. After the above, the Introduction will be (and still is) a bit small and a deeper presentation of the problem statement is very much desirable. Also, it was unclear to me what the significance of this study is. Do you intend to address and contribute to the solution of engineering problems related to corrosion or something else? Please clarify and highlight the significance of the research.

Response:
Our paper conducted the artificial hydrothermal crystallization of spinel; however, the objective is to verify hydrothermal chromian spinel formation in natural hydrothermal environments. To clarify the objective and the significance of our work, we modified paragraph 1 and combined it with paragraph 2. Sentences related to engineering problems were omitted because it made the readers confused. Paragraph 3 was omitted in accordance with your comment. First sentence of the abstract was also omitted.

Revised introduction

Chromian spinel has been recognized as a typical high-temperature igneous mineral recording magmatic process in nature [1–6]. Recent petrological investigations have suggested that chromian spinel could be nucleated from hydrothermal fluids during metamorphic and/or metasomatic events at lower temperature [7–9]. On the other hand, it also has been interpreted as relics of igneous phases [10]. Existence of hydrothermal chromian spinel formation in nature is therefore still controversial, while it would affect previous interpretation of the igneous rocks. Experimental studies can verify the feasibility of hydrothermal chromian spinel in nature. Chromian spinel formation under hydrothermal conditions (ca. <300 °C) has been performed using a batch-type pressure vessel in a pioneering experiment of engineering field [11]; however, such closed system equipment cannot demonstrate chromian spinel formation in a natural oxygen-free hydrothermal system with fluid-circulation. When assuming the formation in natural hydrothermal system, most possible precursors for chromian spinel are Fe-Cr hydroxides because Fe2+(aq) in hydrothermal fluid is oxidized when mixed with dissolved Cr6+ by the difference of the redox potential, resulting in precipitation of Fe-Cr hydroxides. Then reduction of Fe-Cr hydroxides is required to form chromian spinel in classic thermodynamical calculation at equilibrium. However, a previous experimental study has indicated that the non-redox transformation of Fe-bearing minerals could occur under H2-rich acidic hydrothermal conditions via dissolution/reprecipitation as following this reaction [12]:

Fe2O3 + Fe2+(aq) + H2O = Fe3O4 + 2H+      (1)

The reaction (1) is reversible and redox-independent replacement of magnetite by hematite has been experimentally investigated and widely observed in natural iron ores in recent studies [13–15]. The finding led us to the idea that chromian spinel can also be formed by similar non-redox acid-base reaction at low-temperature hydrothermal conditions. In this study, we have developed a high-pressure flow-through reactor to conduct hydrothermal chromian spinel synthesis from Fe-Cr hydroxides with an aqueous ferrous iron (Fe2+(aq)) supply under mildly acidic conditions (pH of 3–5) with the assumption that the conditions of oxygen-free hydrothermal environments. The flow-through experiment was performed at various solid-liquid ratios, initial pHs of the supplied solutions and flow rates to examine the physicochemical conditions, especially lower limit of temperature feasible for chromian spinel formation at 150, 170 and 200 °C.

  1. In Table 1, a brief description of how the experimental conditions were selected would be highly desirable. A minor point is that in my opinion the table needs some rectifications and to tidy up the headers.

Response:
According to the comment, we added the following sentences in Materials and Methods:

Initial pH of HNO3 solution and temperature were set at 2.97–4.00 and 150–200 °C, respectively, assuming mild acidic hydrothermal conditions. Fe-Cr hydroxides were basically synthesized at the Cr/Fe ratio of 0.25 according to the method by [16]. Synthesis of homogeneous Cr-rich starting material was difficult due to the rapid precipitation of Cr hydroxides under atmospheric condition; therefore, we varied the Cr/Fe ratios in the starting materials in the narrow range from 0.2–0.33. The flow rate was set to 1–2 cm3/min to minimize the temperature difference within the cells (<10 °C). The experimental du-rations were 2–7 days.

We also added some rectifications to the Table 1 and tidied up the headers.

  1. Chapter 3.3 is very important but I have the impression that more citations are required to substantiate the interpretations. For example, an important issue is that the formation of spinel after goethite requires reducing conditions. Any comment about that? Is there any major change in terms of dissolved oxygen (or oxygen fugacity?) during the experiments? The recommended reactions are acceptable but is there any further literature supporting them? It would be good to cite more references on the formation of spinels after oxides. Reactions 6 and 7 are not very well supported.

Response:

Recent studies indicated that spinel (magnetite) formation from goethite or hematite could occur redox-independently, and the reaction is also used for Fe2+ supply from the cell 1 in our flow-through system. We added sentences for the explanation and reference in the introduction and discussion part.

Additional sentences in discussion 3.3:

Synthesized Cr spinel consisted of nanosized columnar particles (Figure 6c), which suggests that chromian spinel is formed from other iron minerals, possibly goethite. As shown in reaction (1), hematite can be redox-independently transformed to magnetite with Fe2+(aq) supply via dissolution/reprecipitation at low temperature reducing hydro-thermal conditions (150 °C) where redox kinetics is sluggish [12]. If it is assumed that a similar transformation could occur in the present experiments, then Cr-oxide (eskolaite) and goethite could also be intermediates according to the following reactions:

  1. Also, the text in lines 336-348 is a bit rushed and requires further elaboration. The statement in lines 344-345 is arbitrary and requires better substantiation.

Response:

According to the comment, we revised the paragraph as follows.

Revised paragraph

Although ferrihydrite is not the stable iron phase under the experimental temperatures, the Mössbauser spectroscopy results indicate that most Fe-Cr hydroxides remained after the experiments. The result suggested the existence of Cr or other experimental conditions improve the stability of Fe-Cr hydroxides. Stability of Fe-Cr hydroxides in acidic oxygen-free solution at 150–200 °C has not been reported in previous studies, however, it may be predicted from the solubility and structural features at ambient temperature. Papassiopi et al. (2014) [29] have reported that mixed Fe0.75Cr0.25(OH)3hydroxides have a similar structure to pure ferrihydrite, and the solubility could be predicted as a solid solution of two hydroxides. Mixed Fe0.75Cr0.25(OH)3 indicated lower solubility than pure Cr(OH)3 in the pH range of the experiment at ambient temperature [29], evoking that solubility of Fe-Cr hydroxides are higher than pure ferrihydrite and Cr addition may increase the solubility of ferrihydrite. When considering structural features of Cr3+(OH)3, Amonette and Rai (1990) [16] suggested that existence of Cr(OH)3 is responsible for the persistence of the non-crystalline state and the inhibition of the con-version to a crystalline phase due to the relative inert coordination sphere of Cr3+ to ligand-exchange reactions. Campbell et al. (2002) [30] reported that the presence of Si also significantly increases the temperature for the transformation from ferrihydrite to hematite due to the suppression of crystal growth along [hk0]. Considering with these previous reports, our result suggests that the Cr addition could increase the stability of ferrihydrite by altering structural features. The result of run #4 indicated that most Fe-Cr hydroxides transformed to hematite at 200 °C (Table 1 and Figure 4e), suggesting that the stabilization is ineffective at > 200 °C.

  1. Finally, after revising the introduction, the conclusions should be revised in a way to respond to the problem statement apart from the major findings.

Response:

According to the comment, we revised conclusions as follows:

Revised conclusions:

Hydrothermal chromian spinel synthesis via non-redox transformation was con-ducted using a flow-through reactor to examine the physicochemical conditions at 150, 170 and 200 °C and 5 MPa under mildly acidic oxygen-free conditions (pH > 3–5). XRD data of the recovered solid samples from reaction Cell 2, where Fe-Cr hydroxide was altered with Fe2+(aq) supplied by acid-leaching of magnetite, indicated that chromian spinel was formed from Fe-Cr hydroxide at 150 and 170 °C together with goethite and hematite. FE-SEM and TEM observations suggest that it is most likely that the goethite aggregates transformed to chromian spinel with redistribution of Cr via dissolution/reprecipitation. Although chromian spinel has been recognized as a high-temperature igneous mineral precipitated from magma in previous studies, the present results indicate that chromian spinel can also be formed in mild to low-temperature hydrothermal systems (ca. < 200 °C). Cr redistribution from goethite to chromian spinel observed in our study requires careful evaluation in the use of Cr#(=Cr/(Cr + Al) atomic ratio) as a petrogenetic indicator.

I suggest replacing the term “granular” (describing the crystal habit of the synthesised spinel) with “octahedral”. In my view, Fig. 6B (lower left) shows clearly the top part of an octahedron!

Response:

According to the comment, we replaced the term “granular” with “octahedral”.

Please replace the term “plate-like” with “platy”, which is the usual term to describe hematite and other crystals with this crystal habit. Moreover, “hexagonal platy” would be even better.

Response:

According to the comment, we replaced the term “plate-like, platy” with “hexagonal platy”.

Please replace {001} with {100} in line 246, which is the typical (and what you show us from your observations) family of faces for this columnar goethite.

Response:

According to the comment, we replaced {001} with {100} in line 246.

Please replace {001} with {111} in line 253, which is the typical family of faces of spinel (octahedral).

Response:

According to the comment, we replaced {001} with {111} in line 253.

Also in my view, Figure 7 is not a phase diagram sensu stricto and I suggest changing this term when you cite it in your text.

Response:

According to the comment, we replaced the term “phase diagram” with “stability field”.

We also found some mistakes in the manuscript and corrected as follows.
1) Daisuke Kawamoto’s address was corrected (Ridaimachi → Ridaicho)
2) Solution sampling was conducted from back pressure valve.

3) Fe2+ in reaction formula was replaced with Fe2+(aq).
4) References related to silica solubility were corrected (Morrey 1951→1962) and added (Fournier and Rowe, 1977) in the list.

Reviewer 3 Report

The manuscript should be accepted in the present form.

Author Response

Dear Reviewer 3,

We greatly appreciate your valuable comments. According to your comments our manuscript was revised as follows.

  1. (Line no. 30) The brief implication of large quantity of Fe-Cr ferrihydrites that were left after the experiments, can be added here.

Response:

According to the comment, we added the brief implication of remained Fe-Cr ferrihydrites in abstract.

  1. (Lines 48-52) Few lines related to the chemical behaviour of chromian spinel in cosmochemistry can also be added to the introduction part for the wholesome idea for the readers.

Response:
Thank you for your suggestion. We understand that meteorites also include chromian spinel which can be utilized for understanding of parent body evolution model. However, other reviewers suggest us to simplify the introduction and implication, so that we revised introduction focusing on natural acidic hydrothermal system related to iron ores.

  1. (Lines 55-62) This section should be the last part of introduction without explaining the much of conclusions and keeping the intrigue until discussion.
  2. (Lines 58-59) The reason of assuming the reduced conditions in hydrothermal and diagenetic environments beneath the seafloor can be explained here in brief.

Response:

According to the comment, we revised the introduction.

Revised introduction

Chromian spinel has been recognized as a typical high-temperature igneous mineral recording magmatic process in nature [1–6]. Recent petrological investigations have suggested that chromian spinel could be nucleated from hydrothermal fluids during metamorphic and/or metasomatic events at lower temperature [7–9]. On the other hand, it also has been interpreted as relics of igneous phases [10]. Existence of hydrothermal chromian spinel formation in nature is therefore still controversial, while it would affect previous interpretation of the igneous rocks. Experimental studies can verify the feasibility of hydrothermal chromian spinel in nature. Chromian spinel formation under hydrothermal conditions (ca. <300 °C) has been performed using a batch-type pressure vessel in a pioneering experiment of engineering field [11]; however, such closed system equipment cannot demonstrate chromian spinel formation in a natural oxygen-free hydrothermal system with fluid-circulation. When assuming the formation in natural hydrothermal system, most possible precursors for chromian spinel are Fe-Cr hydroxides because Fe2+(aq) in hydrothermal fluid is oxidized when mixed with dissolved Cr6+ by the difference of the redox potential, resulting in precipitation of Fe-Cr hydroxides. Then reduction of Fe-Cr hydroxides is required to form chromian spinel in classic thermodynamical calculation at equilibrium. However, a previous experimental study has indicated that the non-redox transformation of Fe-bearing minerals could occur under H2-rich acidic hydrothermal conditions via dissolution/reprecipitation as following this reaction [12]:

Fe2O3 + Fe2+(aq) + H2O = Fe3O4 + 2H+      (1)

The reaction (1) is reversible and redox-independent replacement of magnetite by hematite has been experimentally investigated and widely observed in natural iron ores in recent studies [13–15]. The finding led us to the idea that chromian spinel can also be formed by similar non-redox acid-base reaction at low-temperature hydrothermal conditions. In this study, we have developed a high-pressure flow-through reactor to conduct hydrothermal chromian spinel synthesis from Fe-Cr hydroxides with an aqueous ferrous iron (Fe2+(aq)) supply under mildly acidic conditions (pH of 3–5) with the assumption that the conditions of oxygen-free hydrothermal environments. The flow-through experiment was performed at various solid-liquid ratios, initial pHs of the supplied solutions and flow rates to examine the physicochemical conditions, especially lower limit of temperature feasible for chromian spinel formation at 150, 170 and 200 °C.

  1. (Lines 100-103) How does amorphous silica solubility help in accuracy of temperature measurement?? Could authors explain this a bit??

Response:

According to the comment, we revised the sentence as follows.

Revised sentence

The accuracy of the temperature measurement was examined by measuring the dissolved amorphous silica concentration (amorphous silica C-200, Wako, Japan) inside Cell 2 before experiments for hydrothermal chromian spinel since the solubility has been well known for the temperature dependency.

  1. (Lines 165-168) Could this error of 9.8° C, between observed and calculated temperatures be reduced any further so that it can help in enhancing the accuracy of the simulations?

Response:

Temperature in the flow-through system was controlled by mantle heater of reaction cells and ribbon heater on lines only. We could not set heaters which stabilize temperature in the system on other parts (e.g., acidic solution in the oxygen-free glove box) to prevent the possible oxidation in the system. As shown in Figure 7, temperature difference does not significantly change the stability field of chromite, thus the error of 9.8° C can be acceptable for our objective.

  1. (Lines no 199- 201) How do authors completely rule out the possibility of losing some amount of Fe2+(aq) during hydrothermal alteration in Cell 2 during simulations? Please explain.

Response:

We do not rule out the possibility of losing some amount of Fe2+(aq) but discuss possibility of Fe2+ passing through the Cell 2 without addition to the Fe-Cr hydroxide alteration (Second paragraph from the end in 3.3).

  1. (Lines no. 229-230) Mössbauser spectrum of spinel-group mineral does not show clear peaks which could be covered by peaks for ferrihydrites. The observation is good but it should be supported with some reference if possible.

Response:

According to the comment, we added the reference related to Mössbauser spectrum of natural ferrihydrite with wide FWHM (Murad et al., 1988).

  1. (Lines 344-345) The change in the stability of ferrihydrite against heating attributed to Cr impurity seems to be a simple speculation. Could authors explain this in details or provide some reference of any such observations from past studies.

Response:

According to the comment, we revised the paragraph as follows:

Revised paragraph
Although ferrihydrite is not the stable iron phase under the experimental temperatures, the Mössbauser spectroscopy results indicate that most Fe-Cr hydroxides remained after the experiments. The result suggested the existence of Cr or other experimental conditions improve the stability of Fe-Cr hydroxides. Stability of Fe-Cr hydroxides in acidic oxygen-free solution at 150–200 °C has not been reported in previous studies, however, it may be predicted from the solubility and structural features at ambient temperature. Papassiopi et al. (2014) [29] have reported that mixed Fe0.75Cr0.25(OH)3hydroxides have a similar structure to pure ferrihydrite, and the solubility could be predicted as a solid solution of two hydroxides. Mixed Fe0.75Cr0.25(OH)3 indicated lower solubility than pure Cr(OH)3 in the pH range of the experiment at ambient temperature [29], evoking that solubility of Fe-Cr hydroxides are higher than pure ferrihydrite and Cr addition may increase the solubility of ferrihydrite. When considering structural features of Cr3+(OH)3, Amonette and Rai (1990) [16] suggested that existence of Cr(OH)3 is responsible for the persistence of the non-crystalline state and the inhibition of the con-version to a crystalline phase due to the relative inert coordination sphere of Cr3+ to ligand-exchange reactions. Campbell et al. (2002) [30] reported that the presence of Si also significantly increases the temperature for the transformation from ferrihydrite to hematite due to the suppression of crystal growth along [hk0]. Considering with these previous reports, our result suggests that the Cr addition could increase the stability of ferrihydrite by altering structural features. The result of run #4 indicated that most Fe-Cr hydroxides transformed to hematite at 200 °C (Table 1 and Figure 4e), suggesting that the stabilization is ineffective at > 200 °C.

  1. (Lines 350-360) The conclusion paragraph seems to be repetitive. I will request authors to present the conclusions in bullet point form.

Response:

According to the comment, we revised conclusions as follows:

Revised conclusions:

Hydrothermal chromian spinel synthesis via non-redox transformation was con-ducted using a flow-through reactor to examine the physicochemical conditions at 150, 170 and 200 °C and 5 MPa under mildly acidic oxygen-free conditions (pH > 3–5). XRD data of the recovered solid samples from reaction Cell 2, where Fe-Cr hydroxide was altered with Fe2+(aq) supplied by acid-leaching of magnetite, indicated that chromian spinel was formed from Fe-Cr hydroxide at 150 and 170 °C together with goethite and hematite. FE-SEM and TEM observations suggest that it is most likely that the goethite aggregates transformed to chromian spinel with redistribution of Cr via dissolution/reprecipitation. Although chromian spinel has been recognized as a high-temperature igneous mineral precipitated from magma in previous studies, the present results indicate that chromian spinel can also be formed in mild to low-temperature hydrothermal systems (ca. < 200 °C). Cr redistribution from goethite to chromian spinel observed in our study requires careful evaluation in the use of Cr#(=Cr/(Cr + Al) atomic ratio) as a petrogenetic indicator.

We also found some mistakes in the manuscript and corrected as follows.
1) Daisuke Kawamoto’s address was corrected (Ridaimachi → Ridaicho)
2) Solution sampling was conducted from back pressure valve.

3) Fe2+ in reaction formula was replaced with Fe2+(aq).
4) References related to silica solubility were corrected (Morrey 1951→1962) and added (Fournier and Rowe, 1977) in the list.

Round 2

Reviewer 1 Report

The author did not make changes as requested.

Author Response

Dear reviewer 1, 

Thank you for support with reviewing our manuscript. The manuscript was further revised by Academic reviewer's comments as follows.

Academic Editor’s comment:
          Dear Authors, I appreciate your work, but I have an observation. Your study shows that the amount of spinel crystal is very low, and that the ratio Cr/Fe is 0.03 to 0.1. This is not the ratio of a stoichiometric chromite where the Cr is double than Fe. So, this is not a chromite but a Cr-bearing magnetite. You can call it a chromian spinel but the difference with chromite is very important. According to the nomenclature approved by IMA and recently published (Bosi et al., 2019; Eur. J. Mineral.) we should take in consideration the dominant species, in this case, it is iron, so the synthetized spinels are magnetite, maghemite or similar.

Response:

We totally agree with the comment and added an explanation to clarify the classification for the synthesized spinel in line 331 to 336 in the revised manuscript as follows:

Hence, the synthesized chromian spinel is classified to Cr-rich magnetite according to the nomenclature and classification by the International Mineralogical Association in [27] due to the dominance of Fe in the stoichiometric ratios of the oxyspinel group. Note that maghemite, Fe3+-bearing oxyspinel group, is difficult to be formed under Fe2+(aq)-supplied conditions, which has been discussed in [12].

Line 30 to 33 in the abstract of the revised version was also mortified to show that the synthesized spinel was Cr-rich magnetite as follows.

The Cr/Fe ratio of the chromian spinel was smaller than that of the bulk Fe-Cr ferrihydrites, and equivalent to Cr-rich magnetite, suggesting a redistribution of Cr during transformation from goethite to synthesized spinel under the hydrothermal conditions.

In addition, following sentence was added in line 391 to 392 in the conclusion of the revised version.

The synthesized chromian spinel is classified as Cr-rich magnetite by the chemical composition.

We also added the reference related to Mössbauser spectroscopy for magnetite and maghemite in line 237 of the revised version for fair evaluations since the Mössbauser spectroscopy for chromite was only cited at that part. In addition, the sentence to explain the obtained Mössbauser spectroscopy was also modified (line 238 in the revised version).

Academic Editor’s comment:

This is obviously important and changes a lot the significance of your work. So, I think you should re-examine your work in such terms. Considering this, all the comparison with natural chromites are meaningless.

Moreover, in natural environment it is rather difficult to have goethite in association with Cr-bearing spinel or chromite. Usually there could be a rim of the so-called ferritchromite (see literature of chromites from ophiolitic bodies as an example) where maghemite, magnetite, hematite are present but there is no goethite. Some authors hypothesised the possibility of the formation of magnetite from a chromite precursor (Della Giusta et al. 2011; Per Mineral.) and Lughi et al. (2020; Cer. Inter.) showed the formation of magnetite lamellae and hematite exsolution by oxidising a synthetic chromite crystal, so why not thinking that the reverse is possible. However, considering that there is no goethite in contact with chromite in natural occurrences and that, even if present the ratio chromite/goethite would be very high I think that your final sentence “Cr redistribution from goethite to chromian spinel observed in our study requires careful evaluation in the use of Cr#(=Cr/(Cr + Al) atomic ratio) as a petrogenetic indicator” is meaningless in such context.

Response:

According to the comment, we omitted implication to the natural chromite as a geochemical indicator in the revised version as follows:

Line 41, while it would affect previous interpretation of the igneous rocks

Line 42, can verify the feasibility of hydrothermal chromian spinel in nature

Line 397, Cr redistribution from goethite to chromian spinel observed in our study requires careful evaluation in the use of Cr#(=Cr/(Cr + Al) atomic ratio) as a petrogenetic indicator

Some authors hypothesised the possibility of the formation of magnetite from a chromite precursor (Della Giusta et al. 2011; Per Mineral.) and Lughi et al. (2020; Cer. Inter.) showed the formation of magnetite lamellae and hematite exsolution by oxidising a synthetic chromite crystal, so why not thinking that the reverse is possible.

Response:

Thank you for the suggestion. We checked Della Giusta et al. (2011; Per Mineral.) and Lughi et al. (2020; Cer. Inter.); however, our experimental system is oxygen-free, so the implication for the transformation from chromite to magnetite by the oxidation is not applicable for our work.

We also modified the following sentence in the Acknowledgements (line 396 to 397 in the revised version):

The authors thank A. Kurishiba and other staff members in the laboratory of Nano-Micro Mate-rial Analysis, Joint-use Facilities, Faculty of Engineering, Hokkaido University for support with FE-SEM and TEM observations, and T. Yamazaki at the Institute of Low Temperature Science, Hokkaido University for support with TEM observations, and F. Chikanda for support with English correction.

Reviewer 2 Report

Well done and congrats on your work!

Best regards

Basilios Tsikouras

Author Response

Thank you for support with reviewing our manuscript. The manuscript was further revised by Academic reviewer's comments as follows.

Academic Editor’s comment:
          Dear Authors, I appreciate your work, but I have an observation. Your study shows that the amount of spinel crystal is very low, and that the ratio Cr/Fe is 0.03 to 0.1. This is not the ratio of a stoichiometric chromite where the Cr is double than Fe. So, this is not a chromite but a Cr-bearing magnetite. You can call it a chromian spinel but the difference with chromite is very important. According to the nomenclature approved by IMA and recently published (Bosi et al., 2019; Eur. J. Mineral.) we should take in consideration the dominant species, in this case, it is iron, so the synthetized spinels are magnetite, maghemite or similar.

Response:

We totally agree with the comment and added an explanation to clarify the classification for the synthesized spinel in line 331 to 336 in the revised manuscript as follows:

Hence, the synthesized chromian spinel is classified to Cr-rich magnetite according to the nomenclature and classification by the International Mineralogical Association in [27] due to the dominance of Fe in the stoichiometric ratios of the oxyspinel group. Note that maghemite, Fe3+-bearing oxyspinel group, is difficult to be formed under Fe2+(aq)-supplied conditions, which has been discussed in [12].

Line 30 to 33 in the abstract of the revised version was also mortified to show that the synthesized spinel was Cr-rich magnetite as follows.

The Cr/Fe ratio of the chromian spinel was smaller than that of the bulk Fe-Cr ferrihydrites, and equivalent to Cr-rich magnetite, suggesting a redistribution of Cr during transformation from goethite to synthesized spinel under the hydrothermal conditions.

In addition, following sentence was added in line 391 to 392 in the conclusion of the revised version.

The synthesized chromian spinel is classified as Cr-rich magnetite by the chemical composition.

We also added the reference related to Mössbauser spectroscopy for magnetite and maghemite in line 237 of the revised version for fair evaluations since the Mössbauser spectroscopy for chromite was only cited at that part. In addition, the sentence to explain the obtained Mössbauser spectroscopy was also modified (line 238 in the revised version).

Academic Editor’s comment:

This is obviously important and changes a lot the significance of your work. So, I think you should re-examine your work in such terms. Considering this, all the comparison with natural chromites are meaningless.

Moreover, in natural environment it is rather difficult to have goethite in association with Cr-bearing spinel or chromite. Usually there could be a rim of the so-called ferritchromite (see literature of chromites from ophiolitic bodies as an example) where maghemite, magnetite, hematite are present but there is no goethite. Some authors hypothesised the possibility of the formation of magnetite from a chromite precursor (Della Giusta et al. 2011; Per Mineral.) and Lughi et al. (2020; Cer. Inter.) showed the formation of magnetite lamellae and hematite exsolution by oxidising a synthetic chromite crystal, so why not thinking that the reverse is possible. However, considering that there is no goethite in contact with chromite in natural occurrences and that, even if present the ratio chromite/goethite would be very high I think that your final sentence “Cr redistribution from goethite to chromian spinel observed in our study requires careful evaluation in the use of Cr#(=Cr/(Cr + Al) atomic ratio) as a petrogenetic indicator” is meaningless in such context.

Response:

According to the comment, we omitted implication to the natural chromite as a geochemical indicator in the revised version as follows:

Line 41, while it would affect previous interpretation of the igneous rocks

Line 42, can verify the feasibility of hydrothermal chromian spinel in nature

Line 397, Cr redistribution from goethite to chromian spinel observed in our study requires careful evaluation in the use of Cr#(=Cr/(Cr + Al) atomic ratio) as a petrogenetic indicator

Some authors hypothesised the possibility of the formation of magnetite from a chromite precursor (Della Giusta et al. 2011; Per Mineral.) and Lughi et al. (2020; Cer. Inter.) showed the formation of magnetite lamellae and hematite exsolution by oxidising a synthetic chromite crystal, so why not thinking that the reverse is possible.

Response:

Thank you for the suggestion. We checked Della Giusta et al. (2011; Per Mineral.) and Lughi et al. (2020; Cer. Inter.); however, our experimental system is oxygen-free, so the implication for the transformation from chromite to magnetite by the oxidation is not applicable for our work.

We also modified the following sentence in the Acknowledgements (line 396 to 397 in the revised version):

The authors thank A. Kurishiba and other staff members in the laboratory of Nano-Micro Mate-rial Analysis, Joint-use Facilities, Faculty of Engineering, Hokkaido University for support with FE-SEM and TEM observations, and T. Yamazaki at the Institute of Low Temperature Science, Hokkaido University for support with TEM observations, and F. Chikanda for support with English correction.